# The Relationship between Physical Activity and Psychosocial Well-Being during and after COVID-19 Lockdown

**DOI:** 10.3390/bs13120986

**Published:** 2023-11-29

**Authors:** Anna Rosa Donizzetti

**Affiliations:** Department of Humanities, University of Naples Federico II, 80133 Naples, Italy; annarosa.donizzetti@unina.it

**Keywords:** physical activity, hedonic well-being, eudaimonic well-being, positive mental health, self-efficacy, risk perception, COVID-19

## Abstract

The COVID-19 pandemic was an extremely critical event that had a major impact on the physical and mental health of the world’s population. The aim of the present work is to investigate whether physical activity plays a protective role in well-being both during the lockdown and afterwards, also assessing the role played by self-efficacy and perception of the risk of contracting COVID-19, as well as past behavior. To this end, two studies were conducted, the first close to lockdown (N = 1061; 76.3% females; mean age = 37.3—range: 18–80) and the second 2 years later (N = 562; 71.0% females; mean age = 33.2—range: 18–76). Descriptive and correlational analyses were performed, along with structural equation modelling. The results confirm the hypothesis of a positive impact of physical activity on emotional well-being alone during lockdown and on more general psychosocial well-being 2 years later. This result showed that physical activity during the pandemic represented an avoidance strategy from the psychological distress that COVID-19 was generating, whereas 2 years later, it represents a proactive strategy aimed at generating a positive mental health condition. These results provide a better understanding of the role physical activity plays in well-being by representing a resource for protecting as well as promoting the mental health of individuals.

## 1. Introduction

Since 1986, the World Health Organization [1] has understood health as an overall state of physical, psychological, and social well-being, inspiring the ‘bio-psycho-social model of health’ [2]. In a multidimensional view of well-being, therefore, these three components should be understood as sub-dimensions of a single construct of ‘psycho-social well-being’. 

It is widely known that regular physical activity alone is sufficient to produce beneficial effects on physical health. In fact, there is a great deal of research that has confirmed that an active lifestyle and a moderate to high level of cardiorespiratory motor activity can reduce risk factors related to various types of chronic diseases [3]. 

With regard, on the other hand, to psychological well-being, for a long time, studies have focused on the negative side of psychological functioning [4], implying that people are mentally healthy when they do not suffer from negative psychological symptoms [5]. In this sense, the regular practice of physical activity contributes to reducing the symptoms of certain psychological disorders, such as anxiety, insomnia, stress, depression, etc., due to the increased oxygenation of organs, including the brain, hence the ability of sports practice to generate effects in improving mood and preventing various disorders, also reducing the risk of depression [6] and anxiety [7]. 

More recently, it has been clarified that psychological health is not limited to the absence of psychological illness but includes the presence of a number of positive aspects such as positive affectivity and having a purpose in life [8,9]. Studies on psychological well-being have taken different directions of investigation over the years. The first was that of Diener and colleagues [10] on ‘subjective well-being’, understood as a positive emotional experience and the presence of feelings of satisfaction with one’s life.

A hedonic view of well-being based predominantly on the emotional dimension related to the presence of positive emotions and the absence of negative emotions has prompted researchers to use the term emotional well-being to refer to hedonia. The second is that of Ryff [11], who focused on ‘psychological well-being’, understood as optimal psychological functioning. Psychological well-being is part of a eudaimonic perspective of well-being, understood as the ability to pursue meaningful goals for oneself and others, and to increase personal skills and autonomy, and social and relational competencies [12]. Finally, the third direction of studies is that of Keyes [13], who developed the concept of ‘social well-being’, overcoming the limits of a perspective primarily centered on the individual and opening the way to a perspective that values the interaction between the individual and the context. In this understanding, social well-being refers to the perception of one’s place in society and to mutual influences. Subsequently, the three dimensions of well-being were included in a broader and more integrated concept of ‘positive mental health’ [14], which considers social, emotional, and psychological well-being as interdependent and equally relevant.

People’s well-being during the COVID-19 pandemic was severely tested by the perception of the risk of contracting COVID-19. Risk perception is the subjective judgement that individuals create and maintain regarding the characteristics, severity, and manner in which a risk is managed. It is a subjective psychological construct that is influenced by cognitive, emotional, social, cultural, and individual variations both between individuals and between countries [15]. The literature has emphasized the role of risk perception in motivating health protection behavior in general [16,17] and especially during pandemics [18]. In Italy, as in the rest of the world, the psychological impact of COVID-19 has been such that the population has been at a high risk of post-traumatic stress disorder [19,20].

To counteract the psychological impact of COVID-19 on well-being, an important role has been played by physical activity. Studies show that it is intrinsic motivation toward physical activity that predicts activity practice [21]. Intrinsic motivation, which is within the literature investigating the relationship between physical activity and well-being, takes the form of a desire to see one’s hedonic needs satisfied [22,23]. Therefore, physical activity impacts people’s emotional well-being, to be understood as a positive balance of pleasant to unpleasant affect and a cognitive appraisal of satisfaction with life in general [14]. Hedonic need satisfaction associated with physical activity refers to the degree to which physical activity promotes an increase in positive emotions and a decrease in negative ones. In other words, hedonically pleasurable physical activities are those that generate feelings of pleasure and enjoyment [24]. Not surprisingly, studies that have investigated the relationship between physical activity and well-being have predominantly used hedonic well-being indices to assess psychological health [25,26,27,28,29]. In this sense, physical activity plays a protective role as it is associated with happiness, positive affectivity, higher levels of self-esteem, greater life satisfaction, and improved quality of life [30]. Participation in different programs of physical activity such as sports, fitness, and outdoor activities has beneficial effects on emotional well-being in at-risk youth [31], as does engaging in strenuous physical activity [32]. In addition to positively influencing psychological and emotional well-being [33], physical activity also has an effect on social well-being by promoting pro-social behaviors through team sports [34]. It is precisely for these reasons that much attention is paid to physical activity programs that involve not only school environments but all community environments throughout the entire life cycle [35].

Despite a wealth of scientific evidence attesting to the beneficial role of physical activity, too many people still do not engage in any activity, preferring a sedentary lifestyle, to the point that Kohl, Craig, and Lambert [36] spoke of a ‘pandemic of physical inactivity’. The researchers sought to understand which factors contribute to regular physical activity and which inhibit it. 

Among the many factors that affect the performance of physical activity, a prominent role is played by self-efficacy [37]. Studies that have investigated the exercise sphere have shown that self-efficacy is the best predictor of sports performance, as it predicts the adoption, adherence, and execution of exercise-related activities. The predictive role of self-efficacy has been widely demonstrated in both adolescent [38,39,40] and young adult [41] and adult [42,43] age groups. Self-efficacy is a key motivational factor for people to have health-promoting behaviors [44]. Individuals with higher levels of self-efficacy engage more frequently in a regular physical activity program than those with lower levels of self-efficacy [45]. Furthermore, those with higher self-efficacy feel much more motivated to exercise more than people with low self-efficacy [46]. Among the sources from which beliefs about one’s self-efficacy originate is direct experience of effective management, i.e., those experiences in which a person actually and effectively copes with a given situation [47]. 

## 2. Physical Activity during and after the Pandemic

Lockdown was chosen in a large number of countries around the world as a measure to contain the COVID-19 pandemic, with varying degrees of restrictions [48]. During the lockdown, it was not possible to engage in physical activity outdoors or in designated facilities, but many chose to continue or begin physical activity during the period of maximum stringency of the containment measures. Studies conducted during this period showed that those who practiced physical activity possessed better physical and mental health [49], also representing a protective factor against COVID-19 [50]. During the COVID-19 pandemic, physical activity was found to be a protective factor against the psychological burden that this situation entailed [51], being associated with a lower risk of developing anxiety, depression, and stress [52,53,54]; better sleep quality; and higher levels of well-being [55]. In contrast, a reduction in physical activity during lockdown was found to be associated with greater discomfort, lower mental health, and a significant increase in weight and food consumption [55,56].

The COVID-19 pandemic has altered individuals’ physical activity habits, with some studies reporting a large increase and others a large decrease in physical activity [55]. Di Sebastiano and colleagues [57] used a national physical activity monitoring app to determine physical activity levels measured with the device among 2338 Canadians to determine changes in physical activity 4 weeks before the pandemic and 6 weeks after the pandemic declaration. Although moderate to vigorous physical activity was maintained, significant decreases in light physical activity were observed. A meta-analysis conducted [58] on 66 studies involving a total of 86,981 participants, ranging in age from 13 to 86 years, showed that the majority of the studies found a decrease in physical activity and an increase in sedentary behavior, regardless of the sub-population considered or the methodology used. Specifically, the weekly time spent on light, moderate, and vigorous physical activity decreased. An exception is the study conducted by Romero-Blanco and colleagues [59] with university students in which there was an increase in weekly physical activity during the lockdown, with a significant increase among women, motivated by the need to control their weight. Furthermore, these studies also investigated the correlation between the practice of physical activity and mental health during lockdown, showing significant increases in anxiety and depression levels [58]. Therefore, the authors of this meta-analysis concluded that the promotion of physical activity during lockdown should be targeted not only at generally sedentary people, but also at those with high levels of physical activity prior to lockdown. 

With the end of the lockdown and the gradual reopening of activities, people began to return to the habits they had before the pandemic and this also affected physical activity. Comparing physical activity during the pandemic and after it, Lombardo and colleagues [60] observed a ‘return to normality’ in terms of sporting activity. Hargreaves and colleagues [61] showed that rigorous physical activity of moderate intensity significantly decreased during but also after the lockdown for those individuals who had been very active before. However, for those individuals who were moderately active before lockdown, vigorous and moderate intensity activity was significantly higher during lockdown and these levels were maintained afterwards. In a longitudinal study conducted during and after the COVID-19 pandemic [62], it was found that although there was a decrease in physical activity in the general population at the beginning of the COVID-19 pandemic, there is great heterogeneity in longitudinal changes in physical activity in individuals. Indeed, the authors showed that more than 62% of people had minimal changes and another 9% increased their physical activity. However, it is important to note that almost 29% of people experienced a reduction in physical activity over the same period. In addition, among the people with minimal changes in physical activity, 12% remained consistently inactive.

## 3. Aims and Hypothesis

Extensive is the tradition of studies that have dealt with the effects of physical activity on people’s mental and physical health, but the focus has mainly been on well-being understood as the absence of illness and thus the absence of symptoms such as anxiety, stress, depression, etc. In this study, however, the focus has been on well-being as “positive mental health,” hypothesizing not only the impact of physical activity on well-being but especially hypothesizing its different impact on different dimensions of well-being depending on the pandemic and post-pandemic phase experienced by study participants. The idea behind the research design was that during the lockdown, people, experiencing a situation of profound uncertainty about the future, devoted themselves to physical activity with the sole aim of obtaining immediate pleasure and avoiding the pain of the situation they were experiencing by carrying out actions aimed at achieving immediate pleasure, thus with a hedonic orientation to well-being. However, more than 2 years after the outbreak of the pandemic, the practice of physical activity should represent a proactive goal-oriented coping strategy, thus responding to a eudaimonic well-being motivation. 

To this end, the following hypotheses were formulated: 

**Hypothesis** **1.**
*In relation to the exercise sessions (moderate) in line with the literature, it was hypothesized that during the lockdown, there was a decrease in the practice of sport by virtue of the restrictions imposed by the government (H1_s1_), while in post lockdown, there was an increase by virtue of the resumption of daily activities (H1_s2_).*


**Hypothesis** **2.**
*Regarding the relationships between the variables considered, for study 1, it was hypothesized that the perceived risk of contracting COVID-19 was negatively correlated with emotional well-being (H2a_s1_). Furthermore, moderate physical activity conducted during lockdown was hypothesized to positively correlate with physical activity conducted prior to lockdown (H2b_s1_), with physical-activity-related self-efficacy (H2c_s1_), and with emotional well-being alone (H2d_s1_). For study 2, it was hypothesized that the perceived risk of contracting COVID-19 is negatively correlated with psychosocial well-being (H2a_s2_), and that physical activity conducted 2 years after lockdown is positively correlated with physical activity conducted during lockdown (H2b_s2_), with self-efficacy related to physical activity (H2c_s2_) and with all dimensions of well-being (H2d_s2_).*


**Hypothesis** **3.**
*Finally, based on the literature review and the more general hypothesis, we constructed an a priori model to be tested. As shown in Figure 1, for both studies, we expected physical-activity-related self-efficacy to be a positive antecedent of both moderate physical activity conducted in the past (H3a_s1-2_) and moderate physical activity conducted in the last period, i.e., during COVID-19 for study 1 and 2 years after COVID-19 for study 2 (H3b_s1-2_). We then hypothesized that in addition to self-efficacy, past behavior (H3c_s1-2_) and searching for fitness videos (H3d_s1-2_) were also positive antecedents of moderate physical activity conducted in the last month. Finally, we hypothesized that COVID-19 risk perception (H3e_s1-2_) was a negative antecedent and moderate physical activity (H3f_s1-2_) a positive antecedent of emotional well-being for study 1 and of psychosocial well-being for study 2.*


## 4. Materials and Methods

### 4.1. Measures

Making use of a quantitative methodology, a questionnaire containing various validated measures was specifically designed. 

Mental Health Continuum-Short Form (MHC-SF) [63]. The MHC-SF consists of 14 items on 6-point scales, ranging from 1 = never to 6 = every day. It measures the degree of emotional well-being (e.g., of item ‘During the past month, how often did you feel happy’; α: study 1 = 0.82; study 2 = 0.86), social well-being (e.g., of item ‘During the past month, how often did you feel that you belonged to a community’; α: study 1 = 0.81; study 2 = 0.80), and psychological well-being (e.g., of item ‘During the past month, how often did you feel that you had warm and trusting relationships with others’; α: study 1 = 0.85; study 2 = 0.88). Cronbach’s alpha for MHC-SF was 0.90 in study 1 and 0.92 in study 2.

Perceived self-efficacy scale related to physical activity (SEPA) [64,65]. The perceived self-efficacy scale related to physical exercise consists of 12 items rated on a 5-point Likert scale, from 1 = not at all to 5 = very much. The scale measures subjects’ perceptions with respect to their ability to succeed in maintaining an intention to engage in regular physical activity despite psychological problems and despite health and external problems. In this work, given the general objectives, reference was made only to the dimension relating to psychological problems (e.g., of item “I can respect my intention to exercise even when I am tired”; α: study 1 = 0.89; study 2 = 0.89). 

COVID-19 Risk Perception Scale (CoRP) [66]. The perception of COVID-19 risk was detected through four items (e.g., of item Are you worried about getting diseased with COVID-19 yourself?). Items were assessed on a five-point Likert scale (1 = strongly worried; 2 = worried; 3 = not sure; 4 = not too worried; 5 = not worried at all). Cronbach’s alpha was 0.78 in study 1 and 0.87 in study 2.

In addition, the participants were asked questions about their habits with respect to media use and physical activity. Specifically, with respect to media use, they were asked about the average number of hours per day they used social media (from 1 h to 10 h or more) and the frequency with which they used the media to search for fitness videos, yoga, etc. (SVF) (from 1 = not at all to 5 = very much). On the other hand, with regard to the practice of physical activity, in the first study, they were asked how often in a week they devoted at least 10 min to moderate physical activity, referring to the last month (MPA_DL), i.e., during the lockdown phase, and how often they performed the same activity before the lockdown (MPA_BL) (1 = never; 2 = not every week; 3 = 1–2 days a week; 4 = 3–4 days a week; 5 = 5–6 days a week; 6 = every day). In the second study, the reference was always to the last month for current activity (MPA_PL), whereas for past behavior, reference was made to the lockdown period (MPA_DL).

Participants were also asked to report general demographic information such as sex and age. 

Studies were approved by the Institutional Review Board of University of Naples Federico II, Department of Humanities (prot. 9/2020). 

### 4.2. Statistical Analysis

Survey data were then entered into SPSS 26.0 databases [67] and M-Plus software (Version 8.10) [68], and checked/verified by project staff for accuracy. 

Cronbach’s alpha was used to calculate the reliability of the scales. For the psychological scale, an internal consistency should be greater than 0.70, even if an alpha between 0.60 and 0.69 would be considered acceptable [69].

Descriptive statistics were used to analyze the characteristics of the respondents and the study variables. To assess the reported changes in behavior, Student’s *t*-test was used for paired samples (*p* < 0.05). Pearson’s correlation coefficient was used to determine the relationships between all the variables (*p* < 0.05). 

Structural relationships were tested using structural equation modelling (SEM). To assess the goodness of fit of the model, we used, as indicated, chi-squared distribution and the degrees of freedom (χ^2^/df ≤ 3), Standardized Root Mean Square Residual (SRMR ≤ 0.09), comparative fit index (CFI > 0.90), and Tucker–Lewis index (TLI > 0.90). If the results of the Root Mean Square Error of Approximation (RMSEA) are ≤0.05, they are considered to be good, and they are considered reasonable if they are ≤0.09. Evaluating multiple fit indices simultaneously is recommended because the different indices assess different aspects of goodness of fit [70,71,72]. Satisfactory models should show consistently good-fitting results on many different indices.

### 4.3. Participants and Procedure of Recruitment

Participants were recruited at two different times. Data from the first study were collected from 5 to 12 May 2020, in the week following the end of the lockdown, using snowball sampling procedures.

Participants consisted of 1061 Italian people (76.3% females, 23.7% males), aged from 18 to 80 years (M = 37.3, SD = 14.13). In total, 55.9% of the participants were residents in southern Italy, 29.6% in the North, and 14.5% in the Centre. Overall, 45.2% were high school graduates, 38.3% were university graduates, 13.3% have a postgraduate degree (master’s and specialization), 3.1% have a secondary school degree, and a marginal 0.1% have a primary school degree. Moreover, 55.2% of respondents were workers, 29.7% were students, 10.8% were unemployed, and 4.3% were retirees. In addition, 41.2% of participants said they were married or cohabiting, 30.7% were single, 20.9% were engaged, and 7.1% were divorced, separated, or widowed. Finally, participants reported that during the lockdown, they were in the home alone (9.1%) or with family members (73.4%), partners (15.3%), or roommates (2.2%).

Data from the second study were collected from 13 October 2022 to 12 January 2023, using snowball sampling procedures. 

There were 562 participants in the second study (71.0% females, 29.0% males) ranging in age from 18 to 76 years (M = 33.2, SD = 15.01). In total, 80.3% of the participants were residents in southern Italy, 14.7% in the North, and 4.5% in the Centre. A total of 55.2% were high school graduates, 30.6% were university graduates, 9.8% have a postgraduate degree (master’s and specialization), 3.9% have a secondary school degree, and a marginal 0.5% have a primary school degree. Moreover, 45.9% of respondents were students, 43.0% were workers, 7.5% were unemployed, and 3.6% were retirees. In addition, 36.7% were single, 30.2% were engaged, 29.4% said they were married or cohabiting, and 3.7% were divorced, separated, or widowed. 

All data were collected with self-report questionnaires using an internet-based survey [73], implemented on the Google Forms platform. The questionnaire link was distributed to college students and posted on numerous Facebook groups dealing with diverse topics (e.g., Coronavirus issues, crafts, sports, etc.). Participation in the study was voluntary and anonymous. Participants gave consent to participate on the first page of the survey, which took approximately 30 min to complete. Participants received no compensation for participating in the study.

## 5. Results

### 5.1. Study 1

#### 5.1.1. Physical Activity before and during Lockdown

Firstly, an attempt was made to understand whether there was a decrease in physical activity during the lockdown compared to before the lockdown (H1). 

Considering the frequency analysis (Table 1), it emerges that those who had a sedentary lifestyle continued to do so (before = 25.0%; after = 25.3%), while the proportion of the active increased considerably, especially those who stated that they practiced moderate physical activity every day (before = 6.2%; after = 10.8%). 

To test whether these increases were statistically significant, a paired-sample *t*-test was conducted, from which significant increases emerged (M = −0.215; SD = 1.56; t = −4.500; gl = 1060; *p* ≤ 0.000). In particular, we went from an average of 2.79 (SD = 1.45) in the pre-lockdown period to an average of 3.00 (SD = 1.64) during the lockdown. 

Furthermore, the participants stated that they used the internet for an average of 4.55 (SD = 2.48) h per day and used it for an average of 2.38 h per day (SD = 1.23) to search for fitness videos.

#### 5.1.2. Correlations

Zero-order correlations between the measures are shown in Table 2. Results indicate that moderate physical activity during the lockdown is positively correlated only with physical activity conducted prior to the lockdown (H2b; *r* = 0.50 **), with physical-activity-related self-efficacy (H2c; *r* = 0.35 **), and with emotional well-being alone (H2d; *r* = 0.09 **). 

Furthermore, the perceived risk of contracting COVID-19 correlated negatively with emotional well-being (H2a; *r* = −0.10 **).

#### 5.1.3. Testing of the Hypothesized Conceptual Model

We used structural equation modelling to test the structural relationships. The hypothesized model for predicting emotional well-being was tested (Figure 2), and the results confirmed our model, with a good fit between the theoretical and the empirical models: χ^2^ (df) = 164.747 (39), *p* ≤ 0.000; χ^2^/df = 4.22; CFI = 0.97; TLI = 0.96; RMSEA = 0.055 [0.047, 0.064]; SRMR = 0.035. As hypothesized, physical-activity-related self-efficacy is a positive antecedent of both moderate physical activities conducted in the past (H3a; *β* = 0.27 **; *p* = 0.000) and physical activity conducted during lockdown (H3b; *β* = 0.15 **; *p* = 0.000). Furthermore, past behavior (H3c; *β* = 0.34 **; *p* = 0.000) and the search for fitness videos during COVID-19 (H3d; *β* = 0.44 **; *p* = 0.000) also appear to be positive antecedents of the moderate physical activity conducted in the last month. Finally, as hypothesized, COVID-19 risk perception (H3e; *β* = −0.10 **; *p* = 0.002) is a negative antecedent and moderate physical activity (H3f; *β* = 0.09 **; *p* = 0.003) a positive antecedent of emotional well-being.

### 5.2. Study 2

#### 5.2.1. The Practice of Physical Activities during the Lockdown and after 2 Years

Considering the frequency analysis (Table 3) of moderate physical activity conducted 2 years after the lockdown with that conducted during the lockdown (H1), an increase in the number of those with a sedentary lifestyle emerges (during the lockdown = 21.4%; 2 years after the lockdown = 26.0%), and the percentage of active persons, especially those who stated that they practiced moderate physical activity every day, decreased (during the lockdown = 6.6%; 2 years after the lockdown = 4.1%).

To test whether these decreases are statistically significant, a paired-sample *t*-test was conducted. Although we went from an average of 2.84 (SD = 1.43) in the lockdown period to an average of 2.75 (SD = 1.44) in the post-lockdown period, these differences were not statistically significant (M = 0.091; SD = 1.55; t = 1.387; gl = 561; *p* = 0.083). 

In addition, the participants stated that they used the internet on average over the past month for 3.71 h per day (SD = 2.08) and used it 2.04 h per day (SD = 1.03) to search for fitness videos.

#### 5.2.2. Correlations

Zero-order correlations between the measures are shown in Table 4. Results indicate that moderate physical activity after lockdown is positively correlated with physical activity during lockdown (H2b; r = 0.41 **); with total (H2d) (r = 0.16 **), emotional (r = 0.16 **), social (r = 0.10 **), and psychological well-being (r = 0.16 **); and with physical-activity-related self-efficacy (H2c; r = 0.15 **). In addition, the perceived risk of contracting COVID-19 is negatively correlated with emotional well-being (H2a; r = −0.10 **).

#### 5.2.3. Testing of the Hypothesized Conceptual Model

We used structural equation modelling to test the structural relationships. The hypothesized model for predicting psychosocial well-being was tested (Figure 3), and the results partially confirm our model, with a good fit between the theoretical and the empirical models: χ^2^ (df) = 125.000 (39), *p* ≤ 0.000; χ^2^/df = 3.21; CFI = 0.96; TLI = 0.94; RMSEA = 0.063 [0.051, 0.075]; SRMR = 0.047. 

As hypothesized, physical-activity-related self-efficacy is a positive antecedent of both moderate physical activities conducted in the past (H3a; *β* = 0.16 **; *p* = 0.000) and physical activity conducted 2 years after COVID-19 (H3b; *β* = 0.09 *; *p* = 0.038). In addition, past behavior (H3c; *β* = 0.38 **; *p* = 0.000) also appears to be a positive antecedent of moderate physical activity conducted in the last month, in contrast to the search for fitness videos (H3d; *β* = 0.07; *p* = 0.061), which is not statistically significant. Finally, COVID-19 risk perception was not a significant antecedent of psychosocial well-being (H3e; *β* = −0.03; *p* = 0.412), which was instead predicted by moderate physical activity (H3f; *β* = 0.15 **; *p* = 0.000).

## 6. Discussion and Conclusions

The tradition of studies that have dealt with the effects of physical activity and the connection between physical activity and well-being is extensive. At the same time, however, the COVID-19 lockdown represented an unprecedented phase of existence, a phase of life in which all scientific knowledge was questioned or at least could not be taken for granted. In light of this, a first study was conducted close to the lockdown and a second study 2 years later. 

The objective was to first check whether there was a decrease in physical activity during lockdown in line with numerous international [58] and national [74,75] data. The results showed, contrary to hypotheses, that the participants in the first study increased their moderate physical activity during the lockdown, in line with other international [59] and national [76] studies, which showed that there was a maintenance or increase in physical activity during this period, and this was especially true for those who already had a moderately active lifestyle before the lockdown [61]. In relation to the second study, in the absence of investigations linking physical activity during the lockdown and that conducted 2 years later, it was assumed that a return to normality meant a resumption of physical activity previously reduced due to the restrictions imposed by the lockdown. Again, our hypotheses were not confirmed because there was a slight decrease, albeit not statistically significant, probably due to the increase in physical activity during the lockdown that the return to daily life did not allow to be maintained due to the reduction in time available. 

We then set out to verify the connection between moderate physical activity, well-being, and the other psychological dimensions considered. Specifically, in line with the hypotheses, moderate physical activity correlates with emotional well-being in the first study and with all dimensions of well-being in the second study. This should be interpreted considering the different role physical activity plays in the well-being of individuals at two profoundly different historical moments, that of lockdown and that near the end of the pandemic state. Thus, as also ascertained with the structural equation model, during the lockdown, physical activity represented a positive emotional experience for participants [10] that served to generate immediate pleasure by diverting attention from COVID-19-related issues and counteracting the negative effect on emotional well-being of perceived risk of contracting the virus. Two years after the outbreak of the pandemic, physical activity rediscovers its broader role as a promoter of positive mental health that is no longer negatively affected by perceptions of the COVID-19 risk as the pandemic emergency has passed. The role of physical-activity-related self-efficacy is then confirmed, which in addition to correlating is also an important antecedent of current and past physical activity. Finally, the use of technology also played an important role during the pandemic as it supported people in carrying out physical activity during the phase in which it was not possible to use fitness experts. 

Although this study showed an increase in physical activity during the lockdown, it must be emphasized that the frequency with which physical activity was practiced varied greatly among the subjects and despite the increase, there remained people who devoted little or no time to this activity. Added to this is the finding of a slight decrease in moderate physical activity after 2 years, which must keep governments’ attention focused on the support initiatives to be implemented to promote active lifestyles. 

The beneficial effects of physical activity on people’s mental and physical health have long been known, but this study had the merit of detecting well-being not as the absence of illness or through one or more generic items related to personal perceptions of well-being but through a scale capable of capturing the multidimensional nature of this construct. This made it possible to ascertain the different weight that physical activity has and has had on well-being in the emergency and postemergency phases. The data collected thus have important implications for the design of interventions to promote healthy behaviors such as active lifestyle adoption. In fact, the interventions that need to be implemented in emergency periods should promote the practice of physical activity with messages and campaigns that leverage not so much the beneficial effects it can have on health as its ability to generate immediate pleasure that is essential in phases when people are particularly plagued by problems and worries, and thus its ability to help people better tolerate the impact of emergency situations. Conversely, in periods of normality, promotional interventions will need to leverage physical activity’s ability to be a promoter of “positive mental health” [14], through increases in skills, personal autonomy, and relational and social competencies.

Therefore, these results underscore the importance of defining increasingly concrete actions that include physical activity not only in school curricula but also in more general healthcare programs, making this activity part of local and global prevention strategies.

### Limitations of the Study and Future Research Lines

While this study has reaffirmed the importance of physical activity by enabling a better understanding of its effects on different dimensions of well-being, it has limitations that only future research can address. 

The first problematic aspect of this study concerns the measurement of physical activity. For the future, it might be desirable to use direct, non-self-reported measures that are inevitably subject to bias.

A second problem concerns the number of study participants and the method of recruitment. The participants, while not few, were recruited almost predominantly through social platforms through a procedure that does not guarantee the generalizability of the results. Therefore, for the future, probability sampling, considering particular categories of subjects and balancing gender and age, should be carried out in order to test for these stratification variables. 

Finally, the proposed work presented the results of two studies conducted at two different times, on different subjects. A longitudinal rather than cross-sectional study would be desirable for the future, which can better account for the evolution of behavioral styles especially when considering the impact of such a traumatic event as a pandemic.

## Figures and Tables

**Figure 1 behavsci-13-00986-f001:**
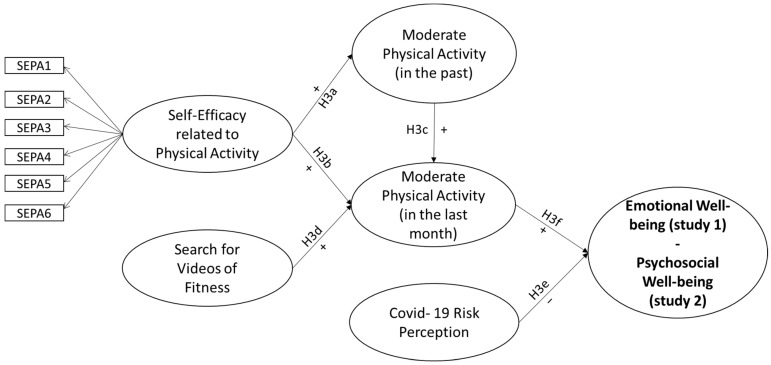
Conceptual model.

**Figure 2 behavsci-13-00986-f002:**
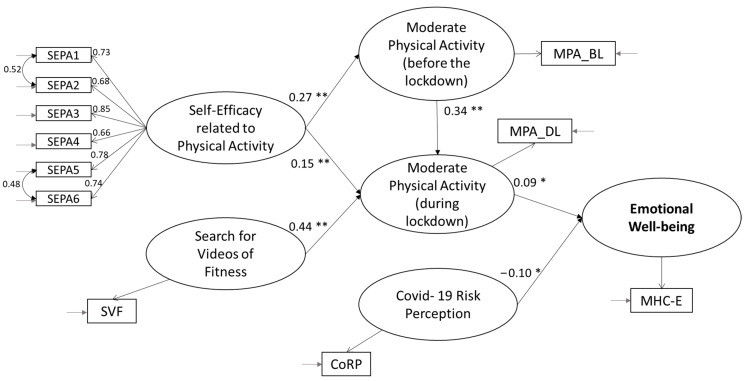
Structural equation model with standardized coefficient estimates (study 1). Notes: ** *p* ≤ 0.000; * *p* ≤ 0.01.

**Figure 3 behavsci-13-00986-f003:**
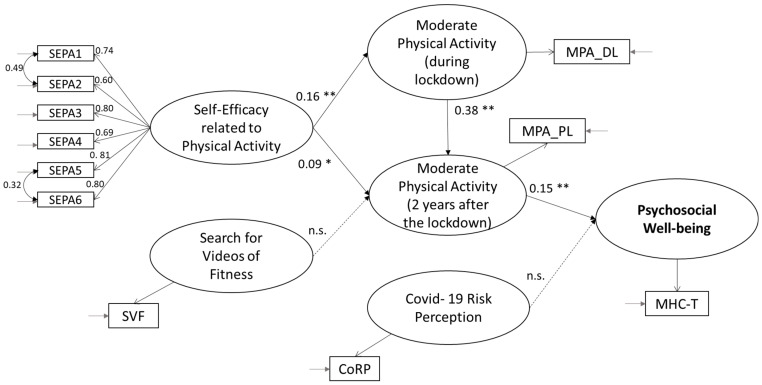
Structural equation model with standardized coefficient estimates (study 2). Notes: ** *p* ≤ 0.000; * *p* ≤ 0.01.

**Table 1 behavsci-13-00986-t001:** Frequency distribution of physical activity, before and during lockdown.

	Moderate Physical Activity	
	Before Lockdown	During Lockdown
Never	25.0%	25.3%
Not every week	19.0%	17.2%
1–2 days per week	25.7%	20.2%
3–4 days per week	18.9%	17.5%
5–6 days per week	5.2%	9.0%
Every day	6.2%	10.8%
Total	100.0%	100.0%

**Table 2 behavsci-13-00986-t002:** Means, Standard Deviations, and correlations between the variables included in the study 1.

	M	SD	1	2	3	4	5	6	7	8
1. Moderate Physical Activity during Lockdown	3.00	1.64	1							
2. Moderate Physical Activity before Lockdown	2.79	1.45	0.50 **	1						
3. Well-Being	3.51	1.00	0.02	0.03	1					
4. Emotional Well-Being	3.75	1.18	0.09 **	0.04	0.79 **	1				
5. Social Well-Being	2.71	1.16	−0.05	−0.02	0.84 **	0.52 **	1			
6. Psychological Well-Being	4.06	1.14	0.03	0.06	0.91 **	0.66 **	0.60 **	1		
7. Self-Efficacy related to Physical Activity	3.63	0.83	0.35 **	0.26 **	−0.07 *	−0.04	−0.11 **	−0.02	1	
8. COVID-19 Risk Perception	3.00	1.64	−0.01	−0.01	−0.06 *	−0.10 **	−0.05	−0.03	0.02	1

Notes: ** *p* < 0.001; * *p* < 0.05.

**Table 3 behavsci-13-00986-t003:** Frequency distribution of physical activity during and after lockdown.

	Moderate Physical Activity	
	During Lockdown	Two Years after the Lockdown
Never	21.4%	26.0%
Not every week	21.2%	21.0%
1–2 days per week	29.2%	21.2%
3–4 days per week	15.4%	20.0%
5–6 days per week	6.2%	7.7%
Every day	6.6%	4.1%
Total	100.0%	100.0%

**Table 4 behavsci-13-00986-t004:** Means, Standard Deviations, and correlations between the variables included in the study 2.

	M	SD	1	2	3	4	5	6	7	8
1. Moderate Physical Activity during Lockdown	2.84	1.43	1							
2. Moderate Physical Activity before Lockdown	2.75	1.44	0.41 **	1						
3. Well-Being	3.43	0.99	0.07	0.16 **	1					
4. Emotional Well-Being	3.74	1.16	0.08	0.16 **	0.87 **	1				
5. Social Well-Being	2.67	1.01	0.04	0.10 *	0.85 **	0.63 **	1			
6. Psychological Well-Being	3.90	1.16	0.07	0.16 **	0.93 **	0.76 **	0.65 **	1		
7. Self-Efficacy related to Physical Activity	2.56	0.92	0.15**	0.15 **	−0.13 **	−0.14 **	0.13 **	0.08 *	1	
8. COVID-19 Risk Perception	2.82	1.00	−0.09*	−0.14 **	−0.06	−0.10 *	0.01	−0.06	−0.05	1

Notes: ** *p* < 0.001; * *p* < 0.05.

## Data Availability

The datasets that support the findings of this study are available from the corresponding author upon reasonable request.

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
