# Peer review of "The Relationship between Physical Activity and Psychosocial Well-Being during and after COVID-19 Lockdown"

_behavsci, 2023, doi:10.3390/bs13120986_

Round 1

Reviewer 1 Report

Comments and Suggestions for Authors

Overall, this paper is a well-documented article describing the relationship between physical activity and psychosocial well-being during and after COVID-19 lockdown. 

The paper provides a wide-ranging study background and the research sample size is very large (n = 1061). However, there are some areas that need additional clarification, support, and formatting, and I believe those refinements would help to strengthen the paper. I list them below.

Introduction

Lines 71-73: Authors focus on the psychological impact of Covid -19 in Italy. It could be interesting if you include more general information, or information of other countries too. 

Lines 75-77: When you describe the link between physical activity, well-being and exercise motivation the justification seems vague and I would expect to see more elaboration. Include some other bibliographic reference, as the article of Esmaelizadeh et al., (2022) about intrinsic and external motivation toward physical activity, to define it clearer. Doi: http://dx.doi.org/10.3389/fpsyg.2022.888758

Lines 115-116: Authors focus again only in Italy, but should try to give a broader perspective.

Aims and hypothesis section is too long and confusing, and can make the reader feel lost. The information of that section can be restructured and summarized in one paragraph. Could you condense some ideas to make it easier for the readers and make it more efficient?

Methods

You mention the Mental Health Continuum-Short Form, the perceived self-efficacy scale related to physical activity and the COVID-19 Risk Perception Scale. Are those questionnaires validated? If they are validated include the reference in the manuscript.

I feel confuse about why you include 2 studies in the manuscript, and specifically the structuration used to present them. I would consider restructuring the participants section and include it in the methods.

Results

In the table 1, the sum of all the percentages from “During lockdown” are 100,1%. Correct also in Table 3.

I appreciate that authors include figure 2 and 3 showing the structural equation model with standardized coefficients estimates. It makes the information very visual and comfortable for the readers.

Discussion

The discussion section contains few information, especially if we compare with the length of the introduction. Results and discussion should be the main sections of the manuscript, so I suggest authors to develop and deepen in the ideas of the conclusion, comparing the results with previous articles published. Are the conclusions consistent with the evidence and arguments presented 
and do they address the main question posed?

References

Authors provided many citations in the introduction, which is good. 

References are appropriate and well cited. Even though, DOI is not included in all references. Try to include in the most as possible.

Author Response

Dear Reviewer,

thank you for your valuable suggestions that helped to significantly improve the work proposed for publication in the journal Behavioral Sciences. Attached I provide detailed responses to your comments. Changes made to the revised manuscript are highlighted in yellow.

Reviewer 2 Report

Comments and Suggestions for Authors

Your work deals with an important topic in the prevention of diseases.

Basic findings on the connection between exercise and mental health are already known and were also investigated during the COVID-19 pandemic.

3 Aims and Hypothesis
Why did you decide to pursue your research question despite the detailed knowledge you presented in the introduction?

What is different about your approach so that new findings on the topic can be expected?

4 Materials and Methods
How exactly did you conduct the online survey? Which online platforms did you use? Which target groups did you address? How exactly did the rapid response system you described work? Did a link have to be forwarded?

I would describe this part of the study in a separate subsection under Methods and remove it from the Results. In addition, at the end of the Discussion, a Strengths and Limitations section of the study should describe the strengths and weaknesses of your recruitment method and what this means for the interpretation of the results.

Author Response

(The authors gave the same response as above.)

Round 2

Reviewer 2 Report

Comments and Suggestions for Authors

You have answered my questions and comments adequately.